# Investigating Visual Perception Impairments through Serious Games and Eye Tracking to Anticipate Handwriting Difficulties

**DOI:** 10.3390/s23041765

**Published:** 2023-02-04

**Authors:** Chiara Piazzalunga, Linda Greta Dui, Cristiano Termine, Marisa Bortolozzo, Matteo Matteucci, Simona Ferrante

**Affiliations:** 1Department of Electronics, Information and Bioengineering, Politecnico di Milano, 20133 Milan, Italy; 2Department of Medicine and Surgery, University of Insubria, 21100 Varese, Italy

**Keywords:** gamification, dysgraphia, eye tracking

## Abstract

Dysgraphia is a learning disability that causes handwritten production below expectations. Its diagnosis is delayed until the completion of handwriting development. To allow a preventive training program, abilities not directly related to handwriting should be evaluated, and one of them is visual perception. To investigate the role of visual perception in handwriting skills, we gamified standard clinical visual perception tests to be played while wearing an eye tracker at three difficulty levels. Then, we identified children at risk of dysgraphia through the means of a handwriting speed test. Five machine learning models were constructed to predict if the child was at risk, using the CatBoost algorithm with Nested Cross-Validation, with combinations of game performance, eye-tracking, and drawing data as predictors. A total of 53 children participated in the study. The machine learning models obtained good results, particularly with game performances as predictors (F1 score: 0.77 train, 0.71 test). SHAP explainer was used to identify the most impactful features. The game reached an excellent usability score (89.4 ± 9.6). These results are promising to suggest a new tool for dysgraphia early screening based on visual perception skills.

## 1. Introduction

Dysgraphia is a learning disability that is associated with an impairment in handwriting [1,2]. According to the Diagnostic and Statistical Manual of Mental Disorders, it is classified as a learning disability when writing skills are unsatisfactory given the person’s age, intelligence, and education [3]. It affects transcription, a set of skills that includes handwriting, typing, and spelling.

In general, it is not easy to find consensus on the prevalence of this disorder [4]. Sometimes, the prevalence of generic nonoptimal fine-motor behavior is cited, while other times reports refer to writing deficiencies. As said by Overvelde and Hulstijn in 2011, “[...] In previous studies, the prevalence of handwriting problems among school-age children has been estimated to vary between 5 and 33%”. These discrepancies may be due to the fact that some subjects present dysgraphic writing only temporarily, while for others it is permanent. In fact, higher incidence rates in primary school are linked with the lessening of symptoms of those students that, with practice, can overcome their initial struggles [5]. However, some studies suggest that improvement cannot be possible without some kind of help [6].

It is important to note that dysgraphia is often present in people also diagnosed with other learning disorders, such as dyslexia or attention deficit disorders [7]. This is related to the fact that they share some underlying causes [8,9], even though there are many cases of dysgraphic patients that are not dyslexic and vice versa, suggesting that, despite sharing some characteristics, these disorders are distinct.

The effects of dysgraphia are manifold. The more obvious and manifest one is illegible handwriting, along with poor management of margins and lines [10,11] and poor spatial organization of the writing sheet [12,13,14]. Another common effect is the difficulty in spelling [2]. Moreover, dysgraphia, like all Specific Learning Disorders, has psychological effects as well. In fact, the strong desire to fit in pushes young people to seek academic achievement. When this is impossible to obtain, psychological distress ensues, triggering a vicious cycle that, in turn, discourages underachievers from making more efforts [15,16]. For children affected by learning disorders, this is particularly hard-hitting, since they encounter difficulties that do not depend on their commitment but have causes outside of their reach. Additionally, the attention on academic performance and the scarce knowledge of some disorders, such as dysgraphia itself, can lead teachers to assume that poor results depend on the child’s laziness or lack of effort [17].

Children spend most of their time in class performing fine-motor activities, handwriting being the predominant one [18]. Because of this, dysgraphic children experience a great amount of stress and self-doubt [19], along with the frustration correlated with the gap between what they want to express and what they manage to write [20]. Some may argue that, with the rise of technology, handwriting will slowly disappear, making dysgraphia less of a problem, but studies show the contrary [21,22]. Additionally, typing at a computer is becoming more and more prevalent, and dysgraphia impacts it too, both because it impairs fine-motor coordination, which may make it difficult to type very fast, and because it is often present together with dysorthography, which makes it difficult to spell words correctly.

The difficulties that stem from dysgraphia are aggravated by the delay with which dysgraphia is diagnosed. In fact, there are a variety of tests that attempt to diagnose dysgraphia [23,24,25], but they can be administered when the student has already learned to write and, since handwriting develops in the first years of primary school, usually reaching a plateau in second grade and then becoming automatic in third grade [26]; this means that a potentially dysgraphic pupil is left undiagnosed for two years, struggling to reach their or her peers’ performance and developing unhealthy habits (wrong grip and bad posture). This can, in the future, make therapy harder.

Apart from the fact that available tests provide a belated diagnosis, they share some other critical issues, listed by Asselborn and colleagues in 2018 [27]. First of all, they are all crafted for a specific age range or alphabet, becoming useless for different demographics. Moreover, their core tasks are very different, causing high variability. Finally, they only test the final product and do not investigate other mechanisms. It is also important to note that, in all of them, the subjective evaluation of the examiner is very prominent, making them less precise on borderline cases.

To make up for these criticalities, there is the need to find early indicators that do not involve writing, in order to screen all pupils before difficulties in handwriting emerge and, if needed, to provide therapies and exercises to strengthen their weaknesses. Several studies have linked dysgraphia to visual perception skills [28,29]. Visual perception is the ability that allows to give meaning to what is seen and to elaborate it to complete tasks [30]. Visual perception impairments are often associated with learning disabilities [31], but they are often difficult to analyze. Some studies have investigated ocular movements, since they have been shown to be heavily related to visual perception [32]. For example, eye movements often are guided by visual memory [33], and they can be used to investigate how a visual search is carried out [34]. This relation suggests the possibility to study visual perception not through tasks that require visual abilities but during the process itself, thanks to ocular movements [35,36].

There are, of course, also traditional testing methods for visual perception, such as the Beery VMI (Beery Visual-Motor Integration) [37], the DTVP-3 (Developmental Test of Visual Perception) [38], and the TVPS-4 (Test of Visual Perceptual Skills) [39], but they pose some challenges because tests are subjectively evaluated, and their results can be influenced from the level of stress of the subject [40]. Consequently, a tool that can provide a stress-free environment in which to test visual perception skills, while being objective, would be useful. Digital tools are nowadays more and more present in schools and in young children’s lives, with some governments, such as the Italian one, recommending the use of technology in the SLD screening process.

A way to avoid the stress associated with testing is the use of serious games. Although it is very common to consider the gaming world a source of distraction, violence, and shallow content, this type of medium can provide novel methods and insights, both in research and rehabilitation [41]. Serious games are games designed for a primary purpose that is not entertainment [42], and, even though their main objective is not fun, players are engaged and motivated to keep going, carrying out the gamified tasks without the boredom or fear that kept them back in the first place. Serious games are very successfully applied to learning, but they are also used for other purposes, such as physical exercise, art, or health [43]. Video games have been used to improve and/or investigate visual attention [44], perception [45], and spatial skills [46,47], eventually using eye-tracking techniques as well [48,49]. Moreover, there are many studies suggesting the possibility to lessen the symptoms of some conditions through video games. This applies to learning disabilities, such as dyslexia [50,51] and dysgraphia [19], as well as a variety of other disabilities [52,53]. However, the kind of video games employed in these studies often belong to the action genre, thus offering no link between the actual task performed during the play-trough and the cognitive processes. In particular, there are no video games that adapt clinical methods such as the aforementioned tests to investigate a particular condition.

Given these premises, the aim of the present work is to present the design and development of a serious game that proposes exercises inspired from gamified versions of visual perception tests’ exercises in order to obtain objective scores that could turn out to be correlated with handwriting development and, thus, be an early indicator of dysgraphia. In addition, the ocular movements of the players have been recorded to extract specific features regarding eye movements and fixations during gameplay. Then, they were analyzed to investigate if ocular strategies have an impact on game performance and if eye-tracking features can predict handwriting proficiency.

## 2. Materials and Methods

### 2.1. Game Design

#### 2.1.1. Game Overview

The serious game developed is a 2D, single-player puzzle game, targeted to children of primary school age. It is developed in Unity 2020.1.6f1 for iPad, using JetBrains Rider 2020.2.3 to edit C# scripts. The application was built and launched using Xcode 12.2. The game can be played with touch inputs, but the intended input device is the Apple Pencil.

The application starts with a screen where the examiner can input the ID of the subject, along with their gender. Then, a brief character and setting introduction ensues. With the characters’ guidance, the player carries out the eye tracker calibration. Then, the actual levels begin. Every type of visual perception exercise is explained in an animated tutorial and must be faced in three difficulty levels: easy, medium, and hard. Once all levels have been completed, the game ends.

A brief description of the game’s story, setting, and characters can be found in Appendix A.

#### 2.1.2. Levels

The proposed games are briefly presented and described in Table 1.

### 2.2. Apparatus

#### 2.2.1. iPad and Apple Pencil

The game was played on an iPad Pro 2018 with a 11” screen. The input pencil used is the Apple Pencil 2.

#### 2.2.2. Eye Tracker

During gameplay, subjects wore the Pupil Core headset from Pupil Labs. It is a lightweight frame (22.7 g), without lenses, with two eye cameras and one world camera. The eye cameras sample with a frequency of 200 Hz, while the world camera samples with frequencies ranging from 30 to 120 Hz, depending on the resolution. The eye tracker comes with a suite of open-source software: Pupil Capture, a real-time application to calibrate and record, and Pupil Player, which performs post hoc visualization and analysis.

#### 2.2.3. System Usability Scale

The System Usability Scale (SUS) (https://www.usabilitest.com/ (accessed on 9 December 2022)) is a set of statements about the game. Each statement can be rated on a scale from 1 to 5, with 1 representing the lowest approval with the statement and 5 representing the highest approval. This scale helps determine if the video game was too complex, difficult to navigate or cumbersome.

#### 2.2.4. Characterization Questionnaire

The characterization questionnaire is a set of questions about age, gender, visual corrections, writing, experience with a tablet, experience with a stylus pen, and experience with an eye tracker. The questions, along with possible answers, are reported in Appendix B.

#### 2.2.5. Custom Satisfaction Questionnaire

Subjects were administered an additional questionnaire with questions about their satisfaction with the game and the setup. The questions were as follows:Was the game fun?Are you satisfied with this experience?Was the Apple Pencil comfortable?Was the Apple Pencil light?Do you prefer the Apple Pencil or a normal pen?

The subject was asked to answer with a score from 1 to 5, with 1 representing the lowest approval and 5 representing the highest approval.

Finally, the subjects were asked some qualitative questions:What did you like the most?What did you like the least?Would you change something?

#### 2.2.6. BVSCO-2

The Batteria per la Valutazione della Scrittura e della Competenza Ortografica [25], meaning Set for the Evaluation of Writing and of Ortographic Competence, is a complete test to evaluate all aspects involved in the development of writing. It provides normative data obtained by testing it on 350+ children. It is divided in three main parts, which evaluate, respectively, orthographic competence, the subject’s ability to produce a written text, and handwriting speed. We only included the last part in the protocol, which is, in turn, divided in three tests administered on lined paper. They are as follows:Writing the letters “le”, in cursive, for a minute;Writing the word “uno”, in cursive or in block, for a minute;Writing number words, in cursive or in block, for a minute.

The test items are to be shown by the examiner. The subject then needs to show that he or she has understood the instructions. Finally, the test is performed.

#### 2.2.7. Protocol

The protocol was compliant with ethics and COVID-19 containment procedures. Details can be found in Appendix C. The subject wore the eye tracker and the examiner adjusted the internal cameras to make sure that the pupil was located with a confidence of at least 80% in every eye and head position. The eye tracker was plugged in the examiner’s computer, where the software Pupil Capture was running. The iPad was placed in front of the subject, and the examiner verified that the screen was visible and centered. The application was launched through Xcode, and the subject was asked to play through the game, keeping their head as still as possible, using only tutorials and dialogue to figure out what to do. At the end of every copying and tracing level, the examiner took a screenshot of the iPad through the dedicated function of Xcode. The subject was asked to fill in the characterization questionnaire, the SUS, and the custom questionnaire. Finally, the subject was asked to carry out the BVSCO-2. For each task, the examiner explained what was requested, demonstrated it and offered the possibility to try it before timing. Then, the task was carried out while timing the execution.

### 2.3. Data Analysis

Data analysis was performed offline along with statistical analysis, both in Python 3.8 (Python Software Foundation, https://www.python.org/, (accessed on 1 February 2023)), which was used for the majority of the analyses, and in R 4.0.1 (https://www.r-project.org/, (accessed on 1 February 2023)), which was used for the extraction of drawing features. Packages used include sklearn, scipy, seaborn, pointpats, catboost, and shap. The significance threshold was set at 5%.

Every subject’s record was composed of the answers to the questionnaires, the BVSCO-2 results, the game data, and the eye-tracking data. Records containing missing values were excluded, and eye-tracking data were cleaned up by setting a minimum confidence of 70%. Through the Pupil Player’s plugins, files containing the gaze positions and the fixations were extracted. The SUS scores were transformed using the guidelines, and a single score was computed for each subject.

Then, exploratory data analysis was conducted, building correlation matrices between game data, BVSCO-2 data, characterization questionnaire’s answers, and the indicators extracted from eye-tracking data through the seaborn package. The impact of gender on the BVSCO-2 performance was computed through a t-test through the scipy package. Moreover, to understand which games were more challenging, the variance of their variables was computed, and this value led to the decision to focus on the analysis of some games instead of others.

The graphemes written in the BVSCO-2 were counted following the guidelines. Each exercise is evaluated by the number of produced graphemes. For the first exercise, the one that included writing the letters “le” in cursive, the graphemes which do not respect the sequence are excluded, while the other exercises include all the graphemes, even if the word is written wrongly. For each of the three exercises, the scores were labeled as under- or overthreshold if they were lower or greater, respectively, than the average score reported in the normative data for the corresponding class minus two standard deviations. Then, based on these results, the subjects were labeled as at-risk or not-at-risk: if a subject was labeled as over-threshold for all of the exercises, they were considered not-at-risk, while they were considered at-risk if they turned out to be under-threshold in at least one exercise.

For drawing levels (copying and tracing), additional parameters were computed [54]:Tract discontinuity.Percentage of discontinuous tracts.Length of tracts, both discontinuous and not.*x* and *y* of centroid.Distance from the center (which was computed as the centroid of all children).Dispersion from the centroid.Dispersion from the start.

Moreover, only for the tracing levels:Length of tracts out of trace.Percentage of points out of trace.

Regarding eye-tracking data, in the games where the screen was separated in quadrants, the gaze positions and the fixations were divided into clusters, corresponding to the panels. Even though the position of the child’s head changed throughout the session, the examiner made sure that it was as still as possible during the single games. Moreover, clustering allowed to divide the gaze positions in regions of interest, even though the offset between the eye tracker’s camera and the tablet varied because of each child’s different height and posture. This was achieved through the K-means clustering algorithm, which divided the screen in two horizontal clusters, then dividing the right one in three further portions. Then, each cluster was labeled as “correct answer” or “wrong answer”.

Starting from this and from raw eye-tracking variables (gaze positions and fixations), some quantitative data were computed:Total number of gaze positions and fixations.Average duration of fixations.Percentage of gaze positions on the target image.Percentage of gaze positions on the correct and on the wrong options.Indecision, computed as the number of times the gaze position bounced between the target image and the options.Indecision, computed as the number of times subsequent fixations bounced between the target image and the options.Dispersion of the gaze positions, computed as the Standard Distance.

These data were scaled for each grade (first, second, and third).

Finally, some classification models were built and fitted to predict if the subject was at risk or not. A variety of algorithms were considered:Gaussian Naive Bayes;Random forest;Support Vector Machine (SVM);CatBoost.

The algorithms were trained using game performance data, eye-tracking data, and drawing data. To evaluate their adequacy, the dataset was split into training and test data, assigning 20% of the dataset to the test split and the rest to the train split. Since the dataset is fairly small, to avoid results which were too dependent on the split between train and test data, the models were fitted 6 times, with a different random split each time. Once identifying the best model by means of accuracy performance, the best one was used in a Nested Cross-Validation algorithm, which is particularly useful when the dataset if fairly small, as in this case [55,56]. Finally, SHAP was used to select features and to compute Shapley values to quantify their impact on the output, in order to assess the importance of the various areas considered (visual perception, eye movements, and drawing) on handwriting proficiency.

## 3. Results

### 3.1. Sample Description

A total of 53 subjects participated in the study. Twenty-seven of them were male and 26 were female (age: 7.84 ± 0.90). All subjects attended primary school: 15 attended first grade, 22 second grade, and 16 third grade.

### 3.2. Exploratory Data Analysis

From correlation matrices, a moderate correlation between the attended grade and the BVSCO-2 score emerged (0.47, *p*-value: 0.0001). We also compared the populations of males and females to check if their performances in the BVSCO-2 were similar. We thus considered the number of tests in which they obtained results over the threshold, which ranges from 0 to 3. Males (M = 2.26, SD = 0.76) had worse performances than females (M = 2.65, SD = 0.56), *t*(53) = −2.14, *p* = 0.04. The same difference was found in the scores of the first (males: M = 29.93, SD = 12.13, females: M = 38.08, SD = 12.71, *t*(53) = −2.39, *p* = 0.02) and third test as well (males: M = 42.04, SD = 17.34, females: M = 54.62, SD = 20.96, *t*(53) = −2.36, *p* = 0.02), while the second one did not reach significance. This is coherent with the fact that, historically, girls perform better than boys.

### 3.3. System Usability Score and Satisfaction Questionnaire

The SUS score obtained was 89.4 ± 9.6. This score is greater than 85, which is the threshold that SUS guidelines set for an excellent result. Conversely, 68 is the minimum score that a system has to reach to be deemed usable, and only two subjects were assigned a lower score. Regarding the satisfaction questionnaire, subjects found the Apple Pencil comfortable (average score: 4.98), light (average score: 4.75), and more desirable than a normal pen (average score: 4.72), suggesting it can be a valid input system for young children. Furthermore, the subjects thought the game was funny (average score: 4.92), and they were satisfied overall (average score: 4.96).

### 3.4. Evaluation of Game Difficulties

The mean values and the variance in the game performance variables (times and errors) were computed to find the levels that were the most challenging and which presented a higher variability across subjects. These games were *Masked form constancy*, with an average time of completion for the medium level of 12.18 ± 35.46 s and of 9.03 ± 69.35 s for the hard level, as well as *Figure ground perception*, with an average time of completion of 8.23 ± 88.71 s. The *Masked form constancy* level also presented some of the highest numbers of errors (average: 0.49 for medium level, 0.52 for hard level). Consequently, these two games were chosen for the analysis of the eye-tracking data.

### 3.5. Eye-Tracking Results

Eye-tracking data were visualized both using scatter plots and scanpaths (paths followed by the eyes) to identify interesting metrics to compute.

For each subject, the gaze positions and the fixations in the *Masked form constancy* levels were clustered into the panels that composed the scene (left, top right, middle right, and bottom right) and plotted. Figure 1 shows an example of this division, superimposed on a screenshot of the game.

Then, scanpaths were plotted. Each points represents either a fixation or a gaze position, and the colors of the points are determined by their timestamps: darker colors represent events that happened earlier than the ones in brighter colors.

Figure 2 shows two examples of scanpaths. Examining these images, it emerged that what differentiated subjects with drastically different performances was the sheer number of fixations and gaze positions, their distribution over the different clusters, and their trajectories, particularly when they went back and forth between clusters.

Regarding the *Figure ground perception* game, clusterization was not useful, as the structure of the levels presented only one big panel. However, it was possible to plot scanpaths and to compare those belonging to subjects with good performance with those who, on the contrary, performed poorly. Some examples can be seen in Figure 3.

Children with worse performances had chaotic scanpaths, while those that performed better had ordered scanpaths, similar to those that originate from reading tasks. Consequently, dispersion was identified as an interesting metric.

### 3.6. Classification

Thirty children were over the threshold in all of the BVSCO-2 exercises and were thus labeled as “not-at-risk”, while 23 children were under the threshold in at least one exercise and were thus deemed “at-risk”, as shown in Figure 4.

Table 2 shows the results obtained by the different classification algorithms in terms of the accuracy obtained on training and test data. The accuracy values are the average of the ones obtained by repeating the analysis six times, with six different random splits between train and test data.

From these results, it is apparent that, although the performance is really good on train data, it drops dramatically on test data. This is due to overfitting, as the dataset is quite small. A five-fold Nested Cross-Validation algorithm with the CatBoost classifier was thus used.

The results of the CatBoost algorithm are reported in Table 3.

This algorithm’s performance remains steady across validation and test data thanks to the Nested Cross-Validation, suggesting the possibility to generalize the prediction on unseen data. Moreover, it is apparent that the best values are found among those of recall, which means that the algorithm can identify correctly the children who are actually at risk, particularly when using game performance data as predictors. Finally, for each CatBoost model, Shapley values were computed to visualize the impact of each feature on the model’s prediction, along with its direction.

In Figure 5, the values for the features selected in the game performance model are reported. The points with a blue hue represent subjects with a low value of that feature, while pink points represent high values. Points on the right of the vertical line represent subjects that were not at risk of dysgraphia (i.e., they were over the threshold in all of the BVSCO-2 games). It is apparent that the games that influenced the output the most were *Visual closure* and *Masked form constancy*, as well as *Figure-ground perception*. The direction of these features is as one could expect, since lower values of times and errors are related to lower risks.

In Figure 6, the values for the features selected in the eye-tracking model are reported. Here, the eye-tracking variables that had the strongest influence on the model were the numbers of couples of subsequent fixations that changed clusters (*n cambi fix*), which can be seen as a measure of indecisiveness. Predictably, higher values of this feature lead to a higher risk. The same can be said for the average duration of fixations (*media fix*). Another interesting result is found in the value of the percentage of gaze positions located in the cluster of the target (*% gaze target*). Higher values of this feature corresponded to a lower risk. It is important to note that, even though this model contains eye-tracking features as predictors, almost half of the most relevant predictor belongs to the game performance domain.

In Figure 7, the values for the features selected in the drawing model are reported. Here, the most influential values were the ones extracted from the hard level of the copying game. In particular, those who needed more time to complete the drawing had better results, while an aggravating factor was represented by the discontinuity of the tract and by the spatial organization of the drawing. In fact, variables regarding the centroid (*copy hard x_centroid* and *copy easy x_centroid*) and the size of the image (*copy medium dispersion_from_center*, which is the dispersion of the points from the center point) seem to suggest that those who are at risk tend to draw smaller images in the left portion of the available space, near the reference image. Another important factor was the time needed to complete the copying level (*Time copying hard*). Those who had better results needed more time to complete the copy, especially in the hard level. Another aspect that had great influence on the model was the discontinuity, which appears in several selected features, both in terms of percentage and of tract length. Especially in the most influential variables (*copy hard tract_discont_perc*, *copy hard discontinuity_length*), lower values were related to lower risks.

In Figure 8, the SHAP explainer output for the model which uses all of the features, except for the eye-tracking ones, as predictors is reported. Here, the most impactful features are the ones related to the *Copying* levels, where children who drew bigger pictures on the right side of the available space performed better, coherent with the drawing model. Moreover, regarding other games, the *Masked form constancy* and *Visual closure* levels were the most relevant.

Finally, in Figure 9, the SHAP explainer output for the overall model is reported. The most impactful features are performance in the *Visual closure* levels, performance in the *Form constancy* levels, indecisiveness computed through eye-tracking variables, and percentage of gaze positions on the target. The only surprising result is the one of Errors form constancy in easy mode, as the trend is apparently the opposite. However, this was the very first level presented to the children, which could lead to lower performances, even in subjects who had good results in the following levels.

## 4. Discussion

Dysgraphia is often belatedly detected due to the fact that its diagnosis is possible only when handwriting is completely developed. Technological tools should thus be employed to support schools in an early screening of this SLD through the analysis of indicators such as visual perception skills and eye movements, providing the possibility to begin the strengthening of impaired abilities as soon as possible. The tool should be gamified and fun to eliminate the stress and anxiety factors that sometimes influence the results of standard tests. With these premises, a serious game was developed, presenting levels based on visual perception tests.

The game was tested on 53 children, attending first, second, and third grade of primary school, who wore an eye tracker during the execution and, at the end, were administered the BVSCO-2 test, a writing speed test. Machine learning models were constructed and fitted using game performance and eye-tracking data as predictors, and the BVSCO-2 results were dichotomized in two classes of risk as the dependent variable. It emerged that game performance data predicted the risk category with good results (F1 score: 0.77 on train data, 0.71 on test data). This suggests that visual perception abilities are linked with writing performance, as shown in previous studies [28,29], and that this relation is maintained through the games. Moreover, the importance and influence of each feature was determined by the SHAP explainer. Interestingly, the levels that appeared to have a heavier impact on the output were the *Masked form constancy* and the *Visual closure* ones. This is in line with what other studies have found [57], although form constancy is usually linked more heavily with reading [58].

Similar results were obtained when using eye-tracking data and drawing data as predictors. Regarding eye tracking, raw data from the most challenging levels were manipulated to extract features that could measure indecisiveness and attention. They were then used as predictors along with the performances in those levels, leading to good results (F1 score: 0.64 on train, 0.65 on test). The fact that these values are slightly worse than the ones obtained when using game performance alone can mean that the relation is more feeble, as, in the literature, eye movements are more often related to reading instead of writing [59]. However, the eye tracker did not always allow to have extremely precise measurements, and we were forced to exclude data with a confidence level under 70%, which shrunk the amount of data available. The SHAP explainer highlighted both eye-tracking features and game performance ones, suggesting a similar importance in the construction of the model. Particularly, the eye-tracking features selected suggest that indecisiveness leads to a higher chance of risk.

The model constructed using drawing data from tracing and copying levels had an F1 score of 0.69 on train data and of 0.62 on test data. The most interesting thing about it is the output of the SHAP explainer, which suggests that the most prevalent factors for risk were the time needed to copy an image, the discontinuity of the tract, and the spatial positioning of the copied image. Children at risk needed less time to copy the image, which could be explained by the fact that, if someone is having difficulty at a task, they may want to finish it as soon as possible, while subjects who do not struggle with drawing may have taken more time to enrich the drawing and add more details. Regarding spatial positioning, children at risk drew smaller images in the lower left portion of the panel, nearer to the reference. In the literature, smaller drawings are associated with subjects that children feel negatively about [60]. In this case, the negative feeling could be associated not on the subject but on the act itself. Another explanation could be given by the fact that isochrony and speed–accuracy tradeoff are not respected in dysgraphic subjects [61], leading at-risk children to draw smaller figures in an attempt to achieve a better result.

A model containing all features except for the eye-tracking ones was constructed as well, taking into consideration the fact that an eye tracker is costly and, so, many schools may not have access to one. This model reached an F1 score of 0.79 on train data and of 0.67 on test data. In this model, the spatial positioning of a copied image was the factor which impacted the model the most, with the direction of the variables confirming the findings of the drawing model. Another important factor was performance in the *Masked form constancy* and *Visual closure* levels, which confirms what emerged from the game performance model.

The final model, constructed using all the features, allowed to see the overall impact of features coming from different areas. The *Visual closure* and *Form constancy* levels’ importance was confirmed, as was the indecisiveness which emerged from the eye tracking. In general, it seems that no area prevailed, neither in a sense of performance (although the drawing model was the worst one), nor in a sense of feature importance. This suggests that all of the aspects considered have an impact on handwriting ability.

In almost all of the models, the best metric obtained was recall (game performance: 0.85 on train, 0.86 on test; eye tracking: 0.66 on train, 0.70 on test; drawing: 0.72 on train, 0.70 on test; game and drawing: 0.82 on train, 0.77 on test; and overall: 0.66 on train, 0.75 on test). In the scope of this study, recall can be considered more important than precision, since the purpose of the developed tool is specifically to identify children who are at risk of dysgraphia to provide early support. Overlooking a case of risk would therefore be more costly than to have a false alarm, as it would not be harmful to administer additional training to children, while missing an at-risk child will lead to the very consequences this study is trying to avoid. It is important to remark that lower precision values are often associated with an increase in the recall values [62].

In particular, the model constructed using game performance variables was clearly the best one. This can mean that drawing and eye-tracking variables are not as impactful on the development of handwriting, but it also means that the game alone could be a useful tool for early screening, since both the eye tracker and the Apple Pencil are costly solutions that may not be available in scholastic environments.

Apart from the predictive efficacy of the games, another important aspect of this study was the assessment of usability of the games, since this tool would be useless if it was not well accepted by children. The SUS score was excellent, and the satisfaction questionnaire obtained great results, both on the items regarding the games and the one regarding the Apple Pencil, which was thus deemed an adequate tool to administer these exercises. In general, the eye tracker was well-received as well, but it posed a few challenges. First of all, some children reported that it was uncomfortable, or that it got too hot. Then, the fact that it was head-mounted generated, in some cases, chaotic data, because children were easily distracted and moved their heads, which sometimes made eye-tracking data unusable. For example, in drawing levels, children got very close to the screen, rendering its data useless. Head-mounted eye tracking is, in fact, used mainly when investigating the unconstrained observation of the world [63], while this setting might have benefited from a display-mounted eye tracker.

Surely, another limitation of this study is the sample size, as it is quite small, especially when compared with the amount of variables analyzed. Since the results are promising, it would be optimal to extend the research on a wider audience and to narrow the scope of the abilities tested, since it emerged that some of them have a stronger impact than others. Finally, the Apple Pencil data should be included in the research, as they can provide quantitative measurements of features that have a strong influence on handwriting, such as pressure or tremor [64].

In conclusion, this tool could predict with good results the risk in handwriting, and it was well received by children, suggesting the possibility not only to employ it as a new tool for dysgraphia early screening but also as a way to investigate the role of specific abilities in the development of handwriting.

## Figures and Tables

**Figure 1 sensors-23-01765-f001:**
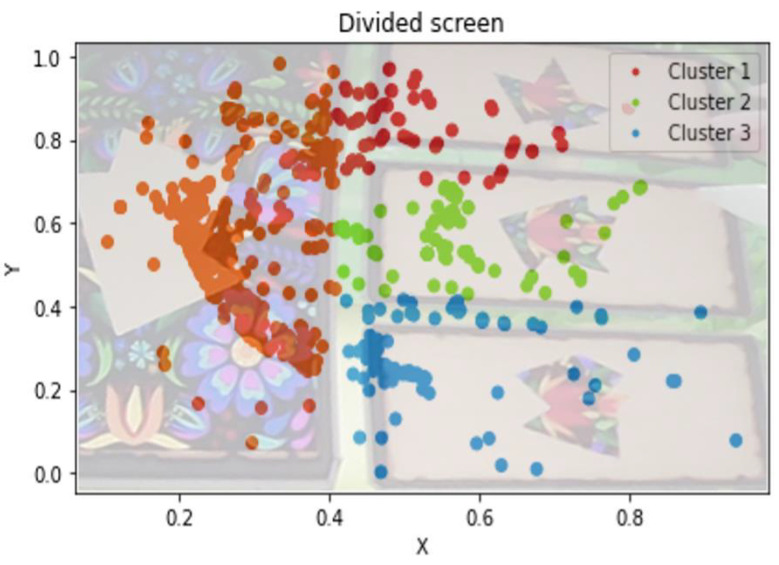
Clustered gaze positions of a subject during the medium level of *Masked form constancy*.

**Figure 2 sensors-23-01765-f002:**
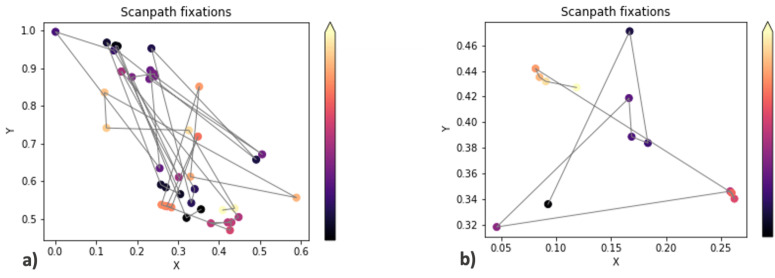
Scanpaths of fixations. Darker colors correspond to fixations that happened earlier than the ones in brighter colors. (**a**) The scanpath of a subject that completed the level in 23.91 s with 3 errors. (**b**) The scanpath of a subject that completed the level in 3.2 s with no errors.

**Figure 3 sensors-23-01765-f003:**
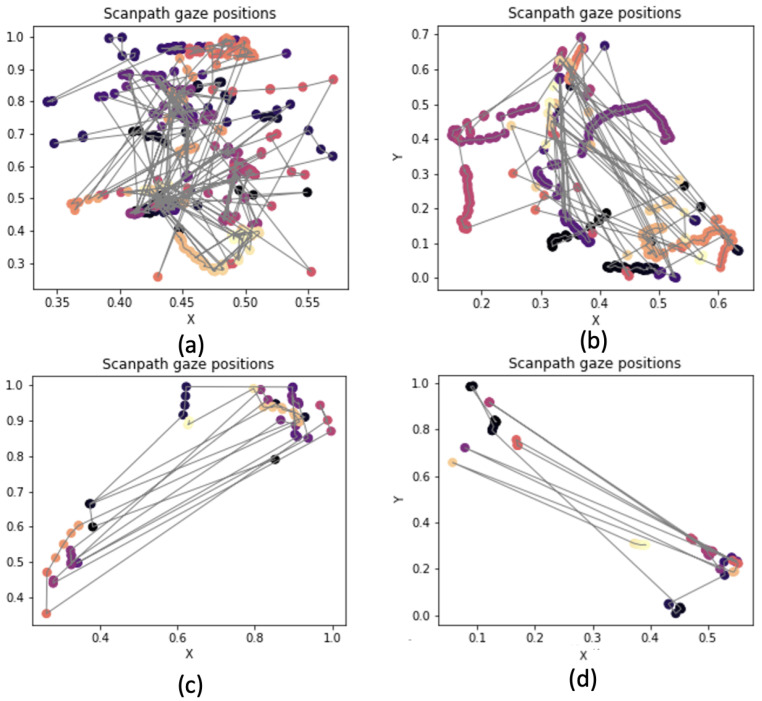
Scanpaths of gaze positions in *Figure ground perception* levels. Darker colors correspond to gaze positions that happened earlier than the ones in brighter colors. (**a**) The scanpath of a subject that completed the level in 9.11 s. (**b**) The scanpath of a subject that completed the level in 7.72 s. (**c**) The scanpath of a subject that completed the level in 3.64 s. (**d**) The scanpath of a subject that completed the level in 4.88 s.

**Figure 4 sensors-23-01765-f004:**
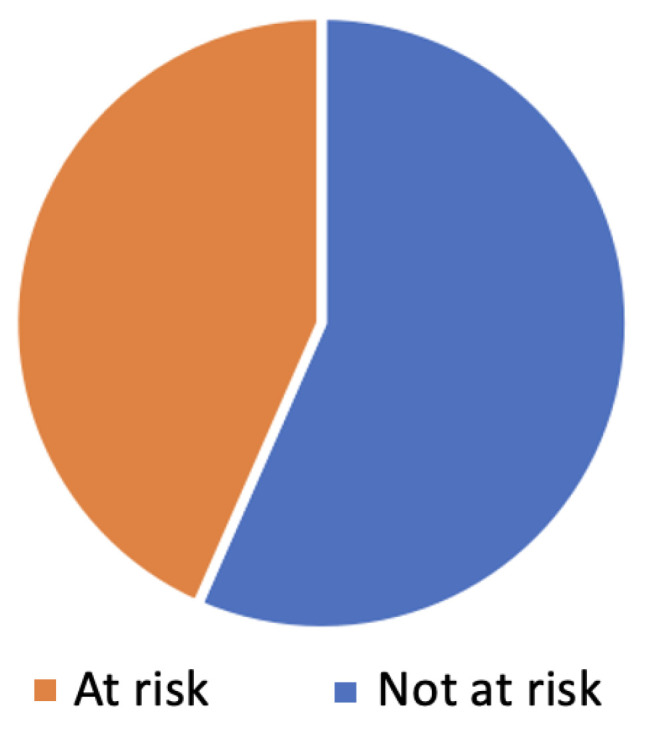
The distribution of children that were at risk or not at risk.

**Figure 5 sensors-23-01765-f005:**
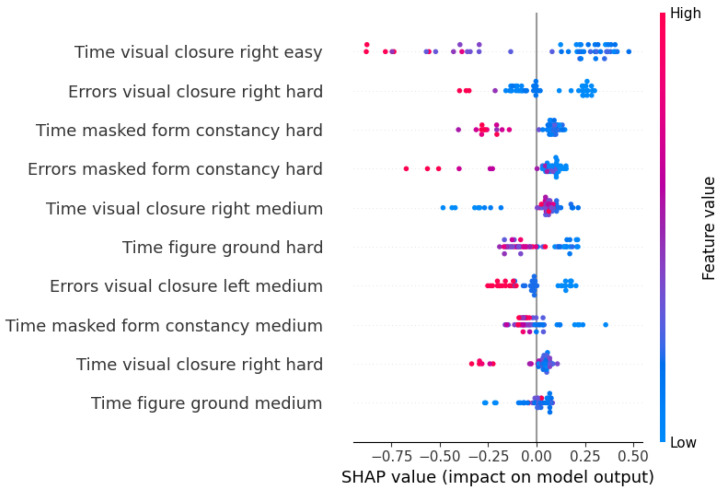
Shapley values of the selected features of the CatBoost model with game performance data as predictors.

**Figure 6 sensors-23-01765-f006:**
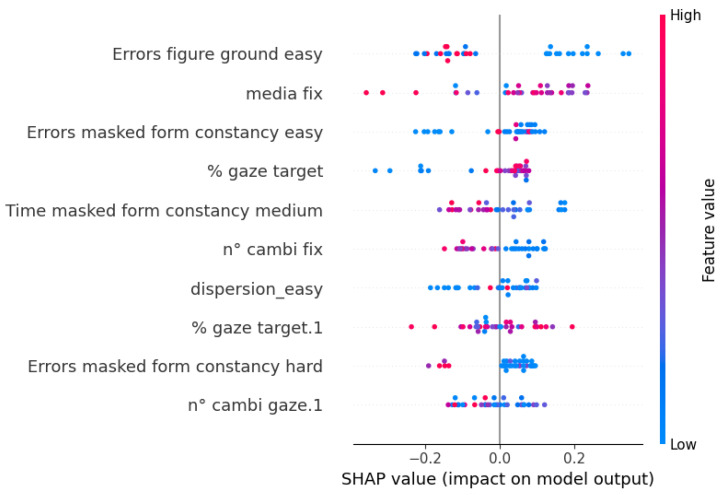
Shapley values of the selected features of the CatBoost model with eye-tracking data as predictors.

**Figure 7 sensors-23-01765-f007:**
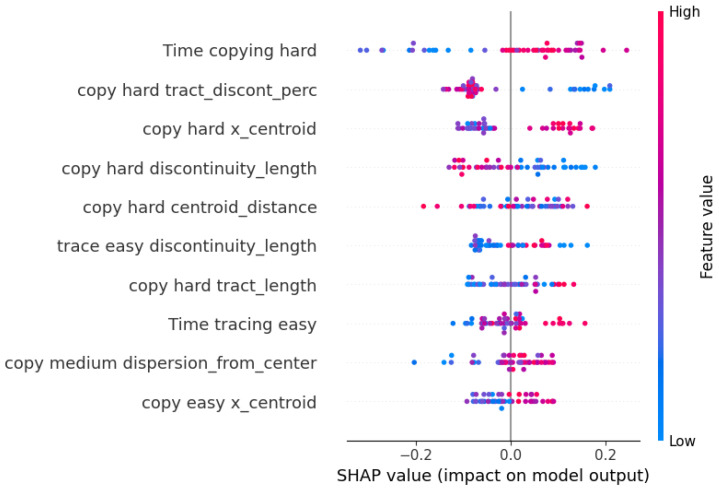
Shapley values of the selected features of the CatBoost model with drawing data as predictors.

**Figure 8 sensors-23-01765-f008:**
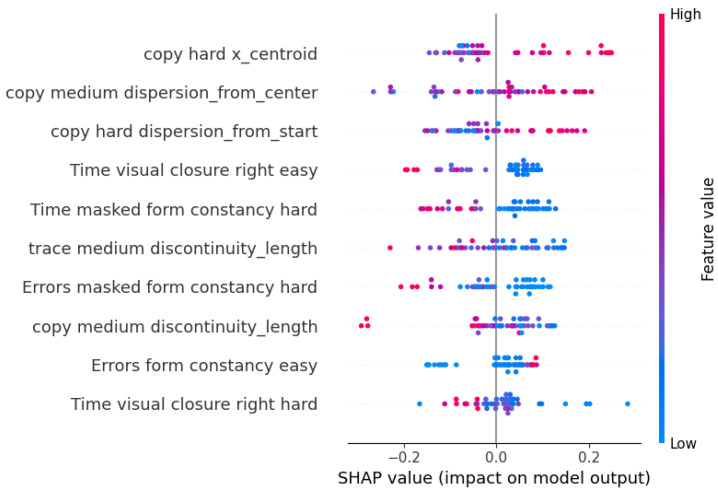
Shapley values of the selected features of the CatBoost model with all of the features, except for eye tracking, as predictors.

**Figure 9 sensors-23-01765-f009:**
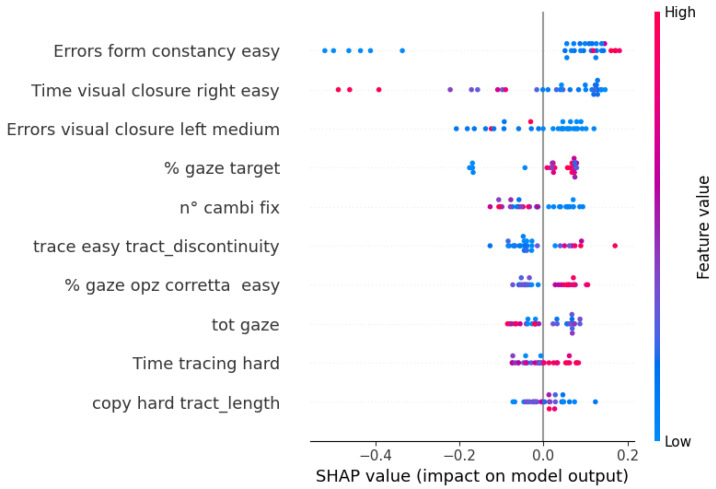
SHAP values of the selected features of the CatBoost model with all of the features as predictors.

**Table 1 sensors-23-01765-t001:** Visual perception games.

Screenshot	Tested Ability	Game Mechanics	Collected Data
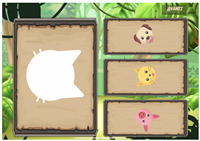	*Form constancy*: the ability to recognize a shape when size, color or orientation changes.	The player must locate the correct shape and drag it to the center of the white outline. When the difficulty rises, the shapes are rotated, thus changing not only color and size but also orientation.	Time to completion Errors
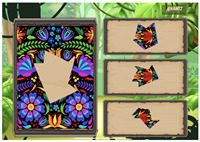	*Masked form constancy*: similar to form constancy, but the shape that has to be recognized is part of a larger image.	The player must locate the correct shape and drag it to the center of the image on the left, which is missing a piece. When the difficulty rises, the options and the target image rotate.	Time to completion Errors
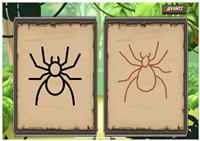	*Copying*: the ability to reproduce an image.	The player must copy the image proposed on the left in the right panel. When the difficulty rises, the reference images become more complex.	Time to completion *x* and *y* of the points of the drawing
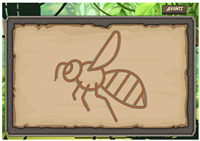	*Tracing*: the ability to reproduce an image on top of a reference.	The player must trace the image proposed, trying to stay inside the line. When the difficulty rises, the reference images become more complex.	Time to completion *x* and *y* of the points of the drawing For every point, a variable indicates if it is inside the trace.
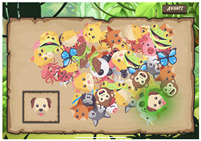	*Figure-ground perception*: the ability to recognize a shape on a confusing background.	The player must find a reference image (shown in the lower left corner) among others. In higher difficulty levels, the number and variety of images increase.	Time to completion Errors
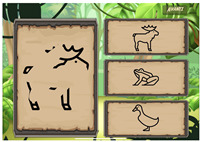	*Visual closure*: the ability to recognize a partially hidden shape.	The player has to recognize and select the image that corresponds to the reference one. The game is presented in two modalities: in the first, the reference is partially hidden; in the second, the options are hidden.	Time to completion Errors

**Table 2 sensors-23-01765-t002:** Averages of accuracy values of different classifiers fitted on train and test data, using game performance, eye-tracking, and drawing features to predict the risk of dysgraphia.

Algorithm	Game Performance (Train)	Game Performance (Test)	Eye Tracking (Train)	Eye Tracking (Test)	Drawing (Train)	Drawing (Test)
Gaussian Naive Bayes	0.86 (0.032)	0.68 (0.069)	0.86 (0.026)	0.55 (0.091)	0.78 (0.058)	0.47 (0.094)
CatBoost	1.00 (0.000)	0.72 (0.121)	1.00 (0.000)	0.43 (0.216)	1.00 (0.000)	0.53 (0.125)
Random Forest	1.00 (0.000)	0.70 (0.115)	1.00 (0.000)	0.43 (0.229)	1.00 (0.000)	0.47 (0.094)
SVM	0.92 (0.040)	0.63 (0.125)	0.95 (0.038)	0.41 (0.160)	0.95 (0.034)	0.52 (0.107)

**Table 3 sensors-23-01765-t003:** Results of the CatBoost algorithm classifier when predicting the risk of dysgraphia using game performance, eye-tracking data, and drawing data in terms of accuracy, precision, recall, and F1 score. The values are the averages and standard deviations of the performances in the five folds. The values on the left are related to the training set, the ones on the right to the test set.

Set of Features	Accuracy (Train)	Precision (Train)	Recall (Train)	F1 Score (Train)	Accuracy (Test)	Precision (Test)	Recall (Test)	F1 Score (Test)
Game performance	0.72 (0.045)	0.73 (0.042)	0.85 (0.061)	0.77 (0.045)	0.61 (0.079)	0.61 (0.036)	0.86 (0.127)	0.71 (0.067)
Eye tracking	0.66 (0.051)	0.74 (0.0.53)	0.66 (0.144)	0.64 (0.090)	0.60 (0.183)	0.63 (0.245)	0.70 (0.215)	0.65 (0.185)
Drawing	0.68 (0.066)	0.72 (0.103)	0.72 (0.109)	0.69 (0.093)	0.55 (0.115)	0.59 (0.118)	0.70 (0.219)	0.62 (0.119)
Game and drawing	0.77 (0.057)	0.80 (0.031)	0.82 (0.092)	0.79 (0.057)	0.59 (0.126)	0.62 (0.074)	0.77 (0.225)	0.67 (0.123)
All of the above	0.70 (0.067)	0.65 (0.125)	0.66 (0.158)	0.64 (0.130)	0.65 (0.112)	0.64 (0.083)	0.75 (0.129)	0.69 (0.093)

## Data Availability

The data presented in this study are available on request from the corresponding author. The data are not publicly available due to privacy reasons.

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
