# Peer review of "Investigating Visual Perception Impairments through Serious Games and Eye Tracking to Anticipate Handwriting Difficulties"

_sensors, 2023, doi:10.3390/s23041765_

Round 1

Reviewer 1 Report

I think this is a well conducted study, but I would have some recommendations for the presentation, which is quite lengthy at times.

In terms of the analysis, I have three main recommendations:

* Repeat splits between training and test and averaging of the results (which will also give you an estimate of the variability)

* Only use the training set (using cross-validation) for determining the final  set of features (this may already be done, but make sure not to "contaminate" the estimates of the test set by using it to determining the feature set

* Fit a model without the eye tracking features (similar to the final model, but leaving the eye tracking data out). The reason is that not every lab will have access to an eye tracker.

For the presentation my main recommendations are:

* Move quite a bit of the details (e.g., the characters in the game, ethics procedure, Covid protocols, questionnaire details) to the appendix.

* Explain early in the paper what the aim of the eye tracking is. On the one hand you seem to be using the eye tracking to better understand how children perform the task, but on the other hand you also use the eye tracking to predict dysgraphia

I was also wondering:

* Will you make the game available?

* Can you make your analysis code available? (gitlab/github/OSF)

* Can you make some of your data (e.g., eye tracking, game performance) available? Anything that will not reveal the identity of the participants. (OSF)

Finally, can you discuss the BVSCO-2, which seems to be the predicted variable, in more detail? How reliable is this measure? Why would you want to replace this measure with the game? (maybe it is in the text, but I missed it).

Please see the comments in the attached PDF for more details.

Author Response

Thank you for your review. We attach the file with the our response.

Reviewer 2 Report

The authors describe a study on the use of an app to assist in identifying issues with handwriting difficulty. Overall the topic will be of interest to many readers (I am not familiar enough with the journal Sensors to know if it is appropriate, but I could imagine a reasonable argument for its appropriateness). In general the study seems to be well designed and explained. Only several minor comments are provided to improve the manuscript.

Line 80: Define these acronyms in the text.

Line 227: Do you really mean writing numbers "in" letters, or perhaps numbers "and" letters?

Line 321: Was sex (male/female) treated as a binary variable for the correlation, and if so is this appropriate?

Figure 4: The figure is missing.

Figure 5: Where the figure says "SHAP" does this refer to "Shapely" as in the previous page?

Author Response

Thank you for your review. You will find the response to your comments in the attached word file. 

Best regards.

Reviewer 3 Report

Dear authors,

Thank you very much for the opportunity to review this article. Having new tools to work handwriting difficulties is a laudable task and I think the manuscript reflects refreshing research on games.

As possible improvements, I indicate the following:

1) 2.2.4. It could go in a table to make it easier to read.

2) I think it would be important for BVSCO-2 to be more detailed.

3) The discussion could further analyze previous studies to compare their results with previous research.

I hope the feedback is constructive, best!

Author Response

Thank you for your review. I attach a word file with my responses. Best regards.

Round 2

Reviewer 1 Report

The authors have addressed all my comments and suggestions. The paper now has a good balance between what is in the main text and the appendix. The machine learning updates also addressed my questions.